# Ionic Liquid—Solidified Floating Organic Drop Microextraction for the Preconcentration of Lead in Environmental Water Samples Prior to Its Determination with Electrothermal Atomic Absorption Spectrometry

**DOI:** 10.3390/molecules29174189

**Published:** 2024-09-04

**Authors:** İlknur Durukan, Barış Yildiz

**Affiliations:** Environmental Engineering Department, Hacettepe University, Beytepe, Ankara 06800, Türkiye; barisyildiz7@hacettepe.edu.tr

**Keywords:** lead, preconcentration, water samples, ionic liquid, electrothermal atomic absorption spectrometry, IL-SFODME

## Abstract

This research investigates the utilization of an ionic liquid combination of solidified floating organic drop micro-extraction (IL-SFODME) to augment the concentration of trace amounts of lead, working as a preliminary stage before electrothermal atomic absorption spectrometry (ETAAS) analysis without the use of chelating agents. Key parameters impacting the microextraction efficiency—including pH, the volume of the ionic liquid (1-Hexyl-3-methylimidazolium hexafluorophosphate, HMIMPF6), temperature, extraction time, and stirring speed—were methodically examined to determine optimal conditions. Under detected optimized conditions, an enhancement factor of 71.2 was obtained for a 15 mL sample solution. The calibration curve exhibited linearity within the concentration range of 0.2–2.5 µg/L, with a detection limit (3σ) of 0.054 µg/L and a quantification limit (10σ) of 0.18 µg/L. For seven replicate measurements of 0.5 µg/L lead, the relative standard deviation (RSD) was ±2.30%. This method was effectively implemented to extract and quantify lead in both reference water and different real water samples, showcasing significantly efficient extraction performance.

## 1. Introduction

Heavy metals are a category of metals and metalloids that possess high densities, atomic weights, or atomic numbers [1]. These elements, such as lead, mercury, cadmium, and arsenic, are characterized by their significant industrial utility and presence in various technological and manufacturing processes [2]. Metals can be originated by natural or anthropogenic processes [3,4]. Heavy metals play crucial roles in numerous sectors, including electronics, metallurgy, agriculture, and energy production [5]. The unique properties of heavy metals, such as high electrical conductivity, malleability, and resistance to corrosion, make them indispensable in the advancement of modern technology and industry [6].

Over the years, industrial growth has led to the extensive consumption of the world’s natural resources for mass production, resulting in significant environmental pollution globally [7]. Numerous organic and inorganic pollutants adversely impact air, water, and soil, thereby threatening the health of humans and other living organisms [8]. Heavy metals, as inorganic pollutants, are particularly concerning because they cannot biodegrade and tend to accumulate in living organisms over time [9]. Among these, lead is a toxic element historically utilized by humans. Recognized for its ductility, corrosion resistance, low conductivity, and softness, lead has been employed in various industries such as paint, construction, mining, ceramics, automotive, and petrochemicals [10,11]. Lead is not biodegradable and tends to accumulate in the body through ingestion, inhalation, or dermal contact, leading to severe health issues, including cognitive impairment, kidney diseases, intellectual disability, and hypertension [12,13]. Consequently, lead, which is challenging to eliminate from the body, poses a significant threat to living organisms. The United States Environmental Protection Agency (USEPA) classifies lead as a priority toxic metal, necessitating careful monitoring and control of its levels [14]. According to the World Health Organization (WHO) and USEPA, the maximum allowable concentrations of lead in drinking water are 0.01 mg/L and 0.015 mg/L, respectively [15,16]. Therefore, it is imperative to monitor and control parameters that directly affect water quality, such as pH, hardness, temperature, dissolved organic matter concentration, and heavy metal concentration, to safeguard both aquatic life and essential water resources [17]. Detecting low levels of lead and other heavy metals is crucial for maintaining the quality and safety of water.

Various analytical techniques have been employed to detect lead from different samples, including Flame Atomic Absorption Spectroscopy (FAAS) [18], Inductively Coupled Plasma-based spectrometry (ICP-OES) [19], and Electrothermal AAS (ETAAS) [20]. Among these methods, ETAAS is preferred due to its advantages over conventional techniques in terms of small reagent consumption, reduced risk of sample contamination, and improved detection limits [21]. However, ETAAS faces challenges such as low sensitivity and signal interference due to the complex matrices of environmental samples. Therefore, a preconcentration step is necessary to separate the analyte from the sample matrix. Several techniques, such as liquid–liquid extraction, dispersive liquid–liquid microextraction [22], cloud point extraction [23], ion exchange [24], co-precipitation [25], and solid phase extraction (SPE) [26], have been extensively utilized for lead preconcentration. Solidified floating organic drop microextraction (SFODME) offers significant advantages due to allowing efficient extraction of target analytes. This technique involves the use of a quite small volume of organic solvent, which floats on the surface of the solution of the sample and also solidifies easily close to room temperature after completion of extraction. The benefits of SFODME include reduced solvent consumption, short and practical sample preparation, and the ability to achieve high preconcentration factors.

The integration of green chemistry practices in laboratory settings marks a significant recent advancement. Green chemistry is characterized by a thorough reexamination of experimental procedures, prioritizing the use of environmentally friendly materials and efficient waste management systems. New methodologies have been developed to evaluate the effectiveness of green chemistry practices compared to traditional techniques. One major challenge is devising sample preparation methods that are both efficient and sustainable. The Green Analytical Chemistry (GAC) approach depends on various factors, including sample collection, preparation, energy consumption, reagents, instrumentation, and overall methodology. To determine the environmental impact of any procedure, a comprehensive evaluation of these factors is essential. Sample preparation is especially critical as it involves the concentration of trace-level analytes and the removal of interfering substances. SFODME stands out due to its requirement for extremely low volumes of extraction solvent. Its advantages include simplicity, cost-effectiveness, minimal use of organic solvents, elimination of chelating agents, and the ability to achieve high enhancement factors, distinguishing it from traditional methods in terms of environmental sustainability and efficiency [27].

This proposed study demonstrates the application of solidified floating organic drop microextraction (SFODME) for the analysis of lead ions in water samples, followed by electrothermal atomic absorption spectrometry (ETAAS) for quantification. In this study, the approach employs 1-Hexyl-3-methylimidazolium hexafluorophosphate (HMIMPF6), an ionic liquid, in lieu of a chelating agent, with 1-dodecanol employed as the extraction solvent. Literature research reveals that this proposed method is the first application of the combination of ionic liquid and SFODME methods without the use of chelating agents for lead extraction. The method is evaluated based on its limit of detection (LOD), limit of quantification (LOQ), and precision, showing excellent performance in these areas. Additionally, it exhibits linearity at trace levels and achieves satisfactory preconcentration. Currently, researchers are focused on developing more environmentally friendly microextraction techniques. This method aligns with green chemistry principles due to its low sample consumption, minimal use of toxic organic solvents and energy, short extraction time, low cost, and satisfactory enhancement factor. Using this approach, lead determination at trace levels, especially in water samples, is effectively achievable with ETAAS.

## 2. Results

### 2.1. Optimization of IL-SFODME Method

#### 2.1.1. Selection of Extraction Solvent

In the IL-SFODME method, choosing an adequate extraction solvent is crucial and needs to fulfill a number of requirements. Firstly, the solvent must possess a density lower than that of water to ensure it remains on top of the aqueous phase. Additionally, it is crucial for the solvent to have a freezing point close to ambient temperature, approximately 25 °C, to enable solidification after the extraction process. To minimize the loss of the organic phase, the solvent should exhibit low volatility and have minimal to no solubility in water. Moreover, the solvent should demonstrate a high efficiency in extracting the target analyte. The solvent should have low toxicity and be used sparingly in accordance with the principles of green chemistry and environmental sustainability. Potential candidates for the extraction solvent include 1-bromohexadecane, 1,10-dichlorodecane, 1-dodecanol, 1-undecanol, 2-dodecanol, and n-hexadecane. Table 1 provides data on the melting points and extraction efficiencies of these solvents. Among these, 1-dodecanol is notable for its extraction efficiency of approximately 98.1% (Table 1). Considering its high extraction efficiency, along with factors such as cost-effectiveness and availability, 1-dodecanol was selected as the extraction solvent.

#### 2.1.2. Optimization of pH

The pH level is the main factor influencing both the formation of metal complexes and their extraction processes. This research examined the influence of pH on the preconcentration of Pb (II) ions across a pH spectrum ranging from 2.1 to 9.0, as depicted in Figure 1. To adjust the pH of the extraction medium, 0.5 mL of various buffer solutions were utilized: 0.2 mol/L KCl and 0.2 mol/L HCl for pH levels between 2.1 and 2.7; 0.1 mol/L potassium hydrogen phthalate and 0.1 mol/L hydrochloric acid for pH values from 3.2 to 4.1; 0.1 mol/L potassium hydrogen phthalate and 0.1 mol/L sodium hydroxide for pH ranges of 5.2 to 6.5; and 0.1 mol/L tris(hydroxymethyl) aminomethane combined with 0.2 mol/L HCl for pH levels from 8.1 to 9.0. The optimal extraction of the complex was observed at pH 2.7, leading to the decision to buffer all subsequent samples to pH 2.7 for further analyses. At pH values below or above the optimal pH of 2.7, the ion-associate-complex (Pb(II)-HMIMPF6) likely becomes charged, hindering its transfer to the organic phase. Given the solubility product constant (Ksp) of lead hydroxide, it is anticipated that at pH values exceeding 8.15, lead precipitates as its hydroxide compound [28].

#### 2.1.3. Optimization of Buffer Amount

The volume of buffer solution used in the extraction process plays a critical role in the efficiency of Pb(II) ion preconcentration. To identify the optimal buffer volume, various amounts were tested: 0.1, 0.2, 0.3, 0.5, 1, and 2 mL (Figure 2). Each buffer volume was assessed for its effectiveness in maintaining the desired pH and ensuring the complete extraction of Pb(II) ions. The experiments revealed that using 0.3 mL of buffer solution yielded the most effective results. This volume consistently maintained the pH at the optimal level of 2.7, facilitating the complete extraction of the Pb(II)-HMIMPF6 complex. Lower volumes, such as 0.1 mL and 0.2 mL, were inadequate in stabilizing the pH, leading to incomplete extraction. Conversely, the use of larger volumes, such as 1 mL, did not lead to a significant enhancement in extraction efficiency and proved to be less practical for the procedure. This is primarily due to this study’s emphasis on minimizing chemical usage to align with green chemistry principles. Consequently, a buffer volume of 0.3 mL was determined to be optimal for the subsequent experiments, thereby ensuring reliable and consistent extraction of Pb(II) ions.

#### 2.1.4. Optimization of Ionic Liquid Amount

The amount of ionic liquid employed in the extraction procedure is a critical factor influencing the effectiveness of Pb(II) ion preconcentration. To determine the ideal volume, various amounts of ionic liquid were evaluated, specifically 2.5, 5, 10, 15, 25, and 50 µL. Each volume was tested for its ability to extract the Pb(II)-HMIMPF6 complex effectively. Experimental findings revealed that the optimal volume for achieving maximum extraction efficiency was 15 µL of ionic liquid (Figure 3). This volume ensured the formation of a stable Pb(II)-HMIMPF6 complex, leading to the complete extraction of Pb(II) ions. Volumes less than 10 µL, such as 2.5 µL and 5 µL, were insufficient for complete complex formation, resulting in incomplete extraction. On the other hand, increasing the volume beyond 15 µL, including 25 µL and 50 µL, did not significantly improve extraction efficiency. Therefore, 15 µL was identified as the optimal volume of ionic liquid, striking the best balance between complex formation and practical efficiency in the extraction process.

#### 2.1.5. Optimization of Extraction Time

To assess the extraction efficiency within a designated timeframe, the transfer of the analyte from aqueous samples to the organic solvent droplet is essential. Consequently, the duration of extraction emerges as a crucial factor that can greatly influence the overall efficiency. A series of experiments were performed under controlled conditions to investigate the impact of extraction time on efficiency, with durations ranging from 5 to 90 min. The results, illustrated in Figure 4, showed that absorbance values increased with longer extraction times until stabilizing at 15 min. Therefore, an extraction time of 15 min was selected as optimal for subsequent experiments.

#### 2.1.6. Optimization of Extraction Temperature

To evaluate the impact of temperature on the extraction of Pb(II) ions using the ionic liquid-supported SFODME technique, experiments were performed over a temperature range from 22 °C to 68 °C with the help of a multi-heater magnetic stirrer (Velp Scientifica Srl, Multi HS-6, Velate MB, Italy). The maximum absorbance, illustrated in Figure 5, was recorded at 26 °C. Beyond this temperature, an increase in solubility of the organic phase led to a decrease in the analytical signal. Consequently, subsequent experimental studies were conducted at 26 °C, which corresponds to room temperature in Ankara, Turkey.

#### 2.1.7. Optimization of Extraction Speed

Agitating the sample solution enhances the speed at which equilibrium is reached between the sample solution and the droplet, facilitating the diffusion of analytes into the organic phase. To investigate this effect, experiments were conducted with stirring rates ranging from 0 to 800 rpm. Higher stirring rates exceeding 650 rpm were noted to decrease absorbance, likely due to splashing and droplet damage. As illustrated in Figure 6, the optimal stirring rate was identified as 650 rpm.

#### 2.1.8. Optimization of Added Dodecanol Amount

In various extraction methodologies, the volume of the solvent used for extraction plays a crucial role in determining the efficiency of the extraction process. The effective transfer of analytes into the solvent microdrop is directly proportional to the surface area of contact between the aqueous phase and the extracting phase. Consequently, an increase in the volume of the drop leads to an enhancement in both the contact area and the overall extraction efficiency. However, further increases in drop volume can cause a negative effect on extraction efficiency. To investigate the impact of extraction solvent volume, different volumes of 1-dodecanol, ranging from 35 to 180 µL, were utilized in the extraction process. The results demonstrated a positive correlation between analyte absorbance and increasing volumes of 1-dodecanol within the range of 35–75 µL, as shown in Figure 7. Over this range, a slight decrease in absorbance was noted between 75 and 100 µL, and for volumes exceeding 100 µL, the absorbance levels remained stable. Therefore, a volume of 75 µL of 1-dodecanol was determined to be the optimal choice for further experimentation.

#### 2.1.9. Optimization of Final Volume

The modified final volume significantly impacts the enhancement factor in the extraction process. High enhancement factors can be achieved by reducing the dilution of the organic phase and the final volume; however, there is a minimum threshold for the final volume for several reasons. Firstly, the ideal final volume for the AAS (Atomic Absorption Spectroscopy, Thermo Fisher Scientific, Waltham, MA, USA) system must accommodate at least three replicate analyses to calculate the standard deviation. Furthermore, it is essential that the viscosity of the final volume is suitable to avoid any obstruction during the suction and injection phases of the autosampling process. Another important factor is to maintain an adequate volume height when the autosampler tip is submerged in the vial, ensuring that a precise quantity of the standard or actual sample is obtained. Experiments were carried out to examine how the final volume of the organic phase influences extraction efficiency, utilizing solutions with varying final volumes between 100 and 350 µL. As expected, the enhancement factors decreased as the preconcentrated solution’s final volume increased. The solution with the highest absorbance value, indicating the greatest extraction efficiency, had a final volume of 175 µL, which was the smallest effective final volume (Figure 8). However, the 100 µL volume had a high standard deviation, likely due to its high viscosity, which negatively affected absorbance values.

### 2.2. Interference Studies

In order to evaluate the feasibility of selectively recovering the analyte amidst the presence of interfering ions, a procedure was conducted utilizing a 15 mL solution that contained 0.75 µg/L of lead, supplemented with foreign ions at different concentration levels. The tolerance limit was defined as the concentration at which the introduced ion caused less than a ±5% relative error in the determination of Pb. Table 2 presents the maximal tolerance limits for the cations and anions. The results indicate that a significant number of the employed ions exhibit negligible impact on the determination of ultra-trace lead in water samples.

### 2.3. Analytical Performance of Proposed Method

The evaluation of the analytical performance of the proposed method was conducted through a series of experiments and calculations. The enhancement factor was calculated by taking the ratio of the slope of the calibration line derived from Pb(II) ion enrichment to the slope of the calibration line obtained from aqueous solutions lacking enrichment. In this investigation, the enhancement factor for Pb(II) ions was determined to be 71.2 (0.2135/0.0030). The method exhibited a linear range of 0.2–2.5 µg/L for lead ions. The limit of detection (LOD), defined as three times the standard deviation (3s) of 10 measurements at the lowest concentration on the calibration plot, was found to be 0.054 µg/L. The limit of quantification (LOQ), representing ten times the standard deviation (10s) of 10 measurements, was determined to be 0.18 µg/L, indicating the lowest concentration that can be reliably quantified. The %relative standard deviation (%RSD) for 0.5 µg/L lead was ±2.30% (n = 7). As detailed in Table 3, the proposed method demonstrates excellent analytical performance.

### 2.4. Accuracy of the Method

To validate the efficacy of the proposed method, recovery experiments were performed using certified reference material. NRC-AQUA-1 (Drinking Water Certified Reference Material for trace metals and other constituents) served as the validation medium, and results are presented as an average of four replicates. Table 4 demonstrates a significant correlation between the results obtained and the established reference values. The recovery rates further validate the precision of the proposed method for measuring lead concentrations.

### 2.5. Analysis of Real Samples

In order to illustrate the practical application of the proposed methodology to actual water samples, lead ions were introduced at designated concentrations into 15 mL of tap water, seawater, and wastewater. Subsequently, the concentrations of lead were quantified utilizing the IL-SFODME technique. The findings regarding the concentration of Pb(II) in these water samples are presented in Table 5.

## 3. Materials and Methods

### 3.1. Reagents and Materials

The Pb(II) solution used in this study was prepared using a stock standard solution (1000 mg/L) of Pb(II) obtained from Certified Reference Material (Certificate #883-02) (Inorganic Ventures, Christiansburg, VA, USA). Working standard solutions were prepared by appropriate dilution of the stock standard solutions. The extraction solvent, 1-dodecanol (CH_3_(CH_2_)_10_CH_2_OH), was sourced from Merck, while 1-Hexyl-3-methylimidazolium hexafluorophosphate (HMIMPF6), an ionic liquid (IL) forming ion pairs with lead ions, was obtained from Sigma-Aldrich, St. Louis, MO, USA). For pH adjustment, a phthalate buffer solution was employed, prepared by combining 0.1 mol/L potassium hydrogen phthalate (Merck, Darmstadt, Germany) and 0.1 mol/L hydrochloric acid (Merck) in suitable proportions. Ethanol (Merck) was used to dilute the organic phase. Matrix modifier chemicals: palladium nitrate (Pd(NO_3_)_2_) (Matrix Modifier, Inorganic Ventures, Inc. 1(800) 669–6799, 10% HNO_3_ 10,000 µg/mL palladium nitrate) and magnesium nitrate hexahydrate (Mg(NO_3_)_2_∙6H_2_O) (Merck) were added in the amount of 0.005 mg Pd + 0.003 mg Mg(NO_3_)_2_ allows a pretreatment temperature of 1000 °C. The appropriate amount of them was injected during the analysis. The concentrations of matrix modifiers were calculated using the recommended conditions obtained from Perkin Elmer. To assess the accuracy of the method, NRC-AQUA-1 Drinking Water Certified Reference Material for trace metals and other constituents, certified reference material from National Research Council Canada (NRC-CNRC), and Beverage Reference Materials were utilized. To ensure the integrity of laboratory materials, all equipment used in the experiments was soaked overnight in a 10% HCl solution and then rinsed three times with deionized water.

### 3.2. Instrumentation

All chemicals used in the experiments were of analytical grade, and solutions were prepared using ultrapure water with a resistance of 18.1 MΩ cm, generated by an Elga Purelab Type-1 ultrapure water device (ELGA LabWater, Lane End, UK). pH measurements were conducted using a Thermo Orion 4-Star pH-conductivity meter (Mettler-Toledo GmbH, Zürich, Switzerland). The sample solutions were heated to the necessary temperature and stirred at the proper speed using a Velp Multi HS-6 multi-heater magnetic stirrer (Velp Scientifica, Usmate Velate, Italy).

Lead concentrations were determined using a Perkin Elmer AAnalyst 800 Atomic Absorption Spectrometer (Spectralab Scientific Inc., Markham, Canada) equipped with a longitudinal Zeeman effect background correction system. A Perkin Elmer electrodeless discharge lamp (EDL) provided radiation at 283.3 nm with a 0.7 nm spectral bandpass. Atomization was performed using a transversely heated graphite tube with an integrated pyrolytic graphite platform. Sample injection was automated using a Perkin-Elmer AS-800 autosampler (Profcontrol GmbH, Schönwalde-Glien, Germany). Argon gas at a flow rate of 250 mL/min served as the inert gas during all stages except for atomization, where the flow was halted. Absorbances were calculated from peak areas obtained during the analysis

### 3.3. Procedure

Aliquots of 15 mL, containing either sample or standard lead solutions, were prepared and transferred into 30 mL wide-neck glass bottles with locking lids. Each bottle was equipped with a stirrer bar for mixing. To adjust the pH to 2.7, 0.3 mL of buffer solution was added to each aliquot. Sequentially, 75 µL of 1-dodecanol extraction solvent and 10 µL of HMIMPF6 ionic liquid were injected into the sample solution using a micro-syringe. The bottles were tightly closed and stirred for 15 min at 650 rpm and 25 °C using a magnetic stirrer. During stirring, the lower-density organic solvent droplet floated on the aqueous surface, facilitating the extraction of Pb(II) ions with the assistance of the ionic liquid.

After extraction, the sealed bottles were cooled in a refrigerator at 4 °C, causing the 1-dodecanol to solidify within approximately 10 min due to its melting point being close to room temperature (24 °C). The solidified droplet was then transferred to micro-vials in the autosampler using a mini spatula, diluted to 175 µL with ethanol in a conical vial, and tightly sealed with parafilm. All standard, real, and modifier samples were loaded into an AS 800 Autosampler (Waltham, MA, USA). In the calibration and determination step, the autosampler sequentially drew 10 µL of Pd, 10 µL of Mg(NO_3_)_2_, and 20 µL of the calibration/sample solutions from separate vials into the tubing and simultaneously injected them into the graphite furnace. The parafilm was removed immediately before sampling. The furnace program detailed in Table 6 was carried out, and the results were recorded, as shown in Figure 9. Each analysis was conducted in triplicate.

## 4. Discussion

The application of solidified floating organic drop micro-extraction (SFODME) combined with 1-Hexyl-3-methylimidazolium hexafluorophosphate ([HMIM][PF6]) as the ionic liquid, without the use of a chelating agent, followed by analysis using ETAAS, represents a novel approach for the determination of Pb in environmental water samples. This method proved highly effective, leveraging [HMIM][PF6]’s advantageous properties, such as low volatility, high thermal stability, and efficient extraction capabilities for Pb. SFODME facilitates practical separation of the preconcentrated lead phase from the aqueous phase due to the easy solidification of 1-dodecanol near room temperature, allowing simple removal of the solidified drop using a spatula.

The IL-SFODME-GFAAS technique demonstrated eye-catching analytical performance, characterized by high sensitivity, a satisfying enhancement factor, and low detection limits, as summarized in Table 7. Comparative analysis with other methods reported in the literature indicates superior or comparable enhancement factors. This method stands out for its operational simplicity, cost-effectiveness, and alignment with principles of green chemistry, particularly minimizing the use of auxiliary substances and energy and reducing derivatives.

Key advantages of this methodology include minimal sample consumption, avoidance of toxic organic solvents, practical extraction duration, affordability, and significant enhancement factor. By enabling lead analysis in water samples, even at trace levels, using conventional ETAAS, this approach enhances the accessibility and applicability of environmental monitoring. Importantly, this study represents the first application of the combined IL-SFODME method without a chelating agent for lead extraction, underscoring its simplicity, speed, and suitability for routine environmental water sample analysis.

## Figures and Tables

**Figure 1 molecules-29-04189-f001:**
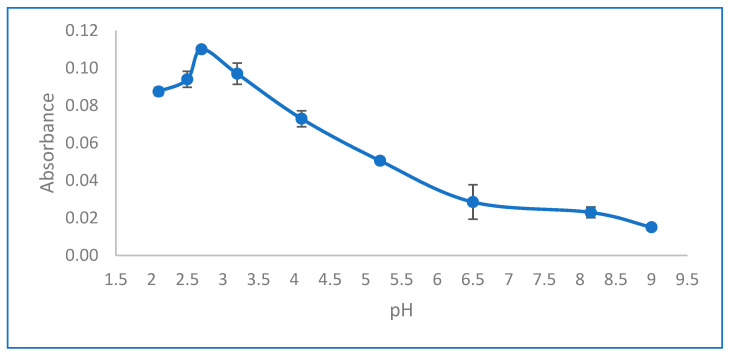
Effect of pH. Conditions: C_0_: 0.75 µg/L Pb(II); sample volume: 15 mL; 0.5 mL buffer solution; ionic liquid amount: 10 µL HMIMPF6; extraction time: 15 min; extraction temperature: 26 °C; extraction speed: 400 rpm; added dodecanol amount: 100 µL; final volume: 200 µL.

**Figure 2 molecules-29-04189-f002:**
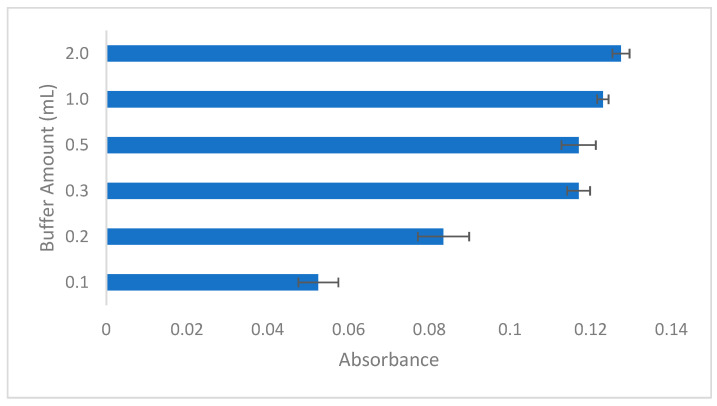
Effect of added amount of buffer. Conditions: C_0_: 0.75 µg/L Pb(II); sample volume: 15 mL; pH: 2.7; ionic liquid amount: 10 µL HMIMPF6; extraction time: 15 min; extraction temperature: 26 °C; extraction speed: 400 rpm; added dodecanol amount: 100 µL; final volume: 200 µL.

**Figure 3 molecules-29-04189-f003:**
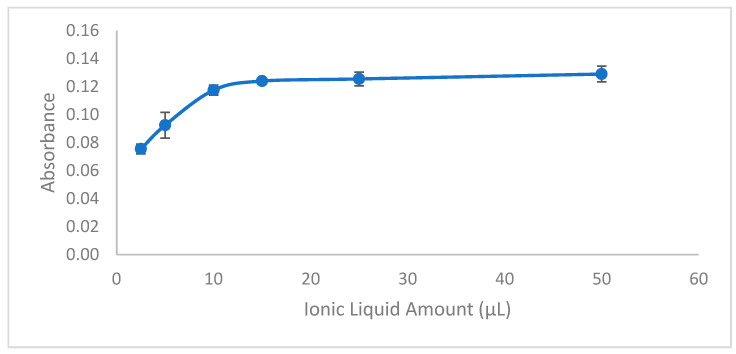
Effect of ionic liquid amount. Conditions: C_0_: 0.75 µg/L Pb(II); sample volume: 15 mL; 0.3 mL buffer pH: 2.7; extraction time: 15 min; extraction temperature: 26 °C; extraction speed: 400 rpm; added dodecanol amount: 100 µL; final volume: 200 µL.

**Figure 4 molecules-29-04189-f004:**
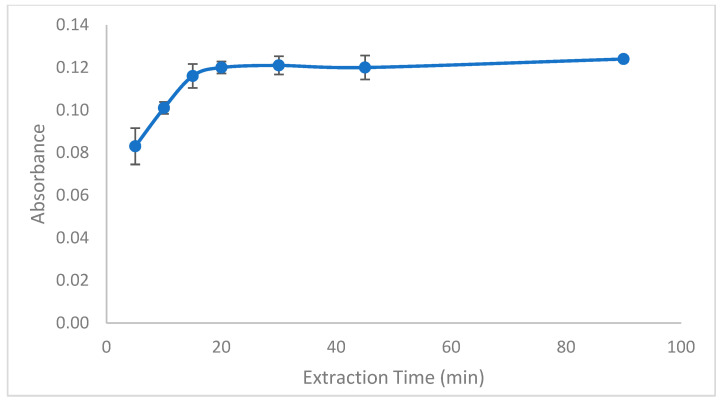
Effect of extraction time. Conditions: C_0_: 0.75 µg/L Pb(II); sample volume: 15 mL; 0.3 mL buffer pH: 0.3; ionic liquid amount: 10 µL; extraction temperature: 26 °C; extraction speed: 400 rpm; added dodecanol amount: 100 µL; final volume: 200 µL.

**Figure 5 molecules-29-04189-f005:**
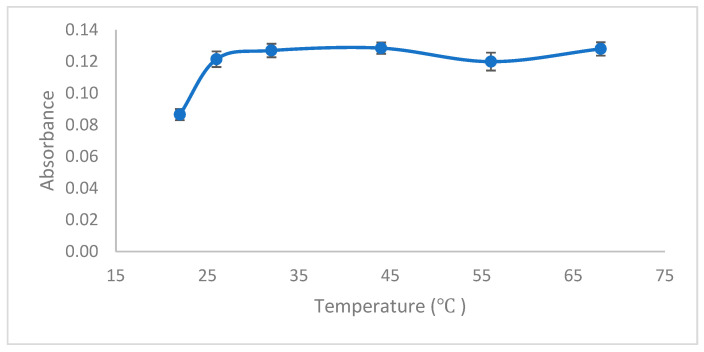
Effect of extraction temperature. Conditions: C_0_: 0.75 µg/L Pb(II); sample volume: 15 mL; 0.3 mL buffer pH: 2.7; ionic liquid amount: 10 µL; extraction time: 15 min; extraction speed: 400 rpm; added dodecanol amount: 100 µL; final volume: 200 µL.

**Figure 6 molecules-29-04189-f006:**
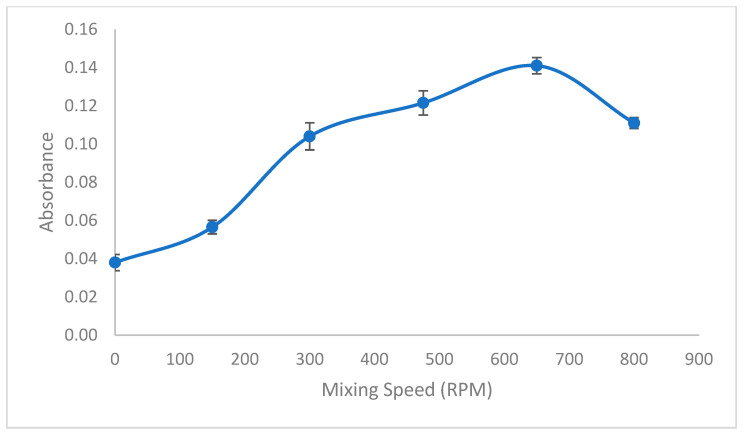
Effect of mixing speed. Conditions: C_0_: 0.75 µg/L Pb(II); sample volume: 15 mL; 0.3 mL buffer pH: 2.7; ionic liquid amount: 10 µL; extraction time: 15 min; extraction temperature: 26 °C; added dodecanol amount: 100 µL; final volume: 200 µL.

**Figure 7 molecules-29-04189-f007:**
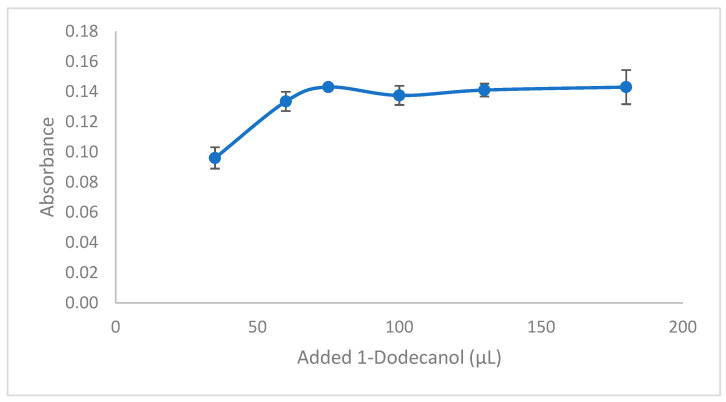
Effect of added extraction solvent. Conditions: C_0_: 0.75 µg/L Pb(II); sample volume: 15 mL; 0.3 mL buffer pH: 2.7; ionic liquid amount: 10 µL; extraction time: 15 min; extraction temperature: 26 °C; mixing speed: 650 rpm; final volume: 200 µL.

**Figure 8 molecules-29-04189-f008:**
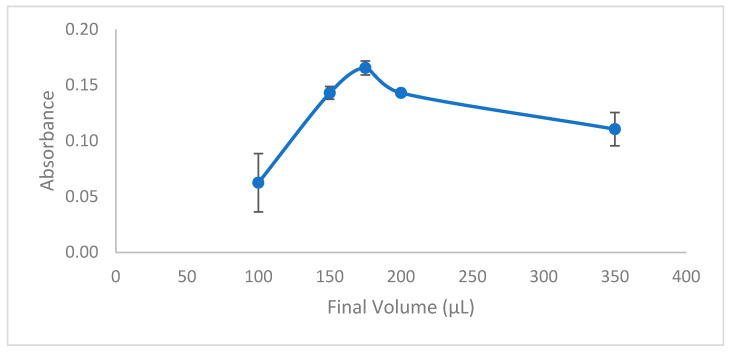
Effect of final volume. Conditions: C_0_: 0.75 µg/L Pb(II); sample volume: 15 mL; 0.3 mL buffer pH: 2.7; ionic liquid amount: 10 µL; extraction time: 15 min; extraction temperature: 26 °C; mixing speed: 650 rpm; added dodecanol amount: 75 µL.

**Figure 9 molecules-29-04189-f009:**
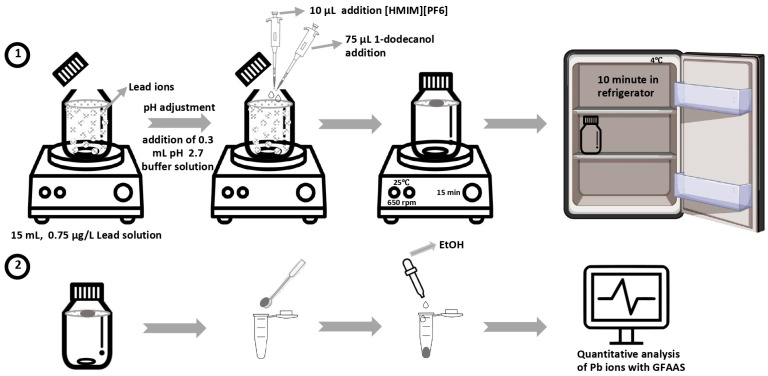
IL-SFODME procedure.

**Table 1 molecules-29-04189-t001:** Melting temperatures and extraction efficiencies of extraction solvents used in SFODME.

Organic Solvent	Melting Point (°C)	Extraction Efficiency of Pb (%)
1-dodecanol	21–24	98.1
2-dodecanol	17–18	97.2
1-undecanol	16	96.5
1-bromohexadecane	17–18	97.1
n-hexadecane	18	95.8
1,10-diclorodecane	14–16	96.2

**Table 2 molecules-29-04189-t002:** Tolerance limits (error < 5%) of diverse ions on the determination of 0.75 µg/L Pb(II) by the proposed method.

Ion	[Pb(II)]/[Ion]	Added As	Ion	[Pb(II)]/[Ion]	Added As
K^+^	>1/5000	KCl	Al^3+^	>1/500	Al(NO_3_)_3_
Na^+^	>1/5000	NaCl	Fe^3+^	1/300	Fe(NO_3_)_3_
Zn^2+^	1/500	Zn(NO_3_)_2_	Cr^3+^	>1/75	Cr(NO_3_)_3_
Ca^2+^	>1/400	CaCO_3_	Co^2+^	1/1000	Co(NO_3_)_2_
Cd^2+^	1/500	Cd(NO_3_)_2_	Se^3+^	1/2000	Na_2_SeO_4_
Hg^2+^	>1/50	HgCl_2_	SCN^−^	1/2000	NH_4_SCN
Mn^2+^	>1/2000	Mn(NO_3_)_2_	Cl^−^	>1/5000	NaCl
Ni^2+^	1/500	Ni(NO_3_)_2_	CO_3_^2−^	1/500	Na_2_(CO_3_)
Mg^2+^	1/300	MgSO_4_	SO_4_^2−^	1/5000	MgSO_4_
Cu^2+^	>1/200	Cu(NO_3_)_2_	CH_3_COO^−^	1/2000	CH_3_COONa

**Table 3 molecules-29-04189-t003:** Analytical performance of the proposed method.

Regression Equation	A = xC + y	0.2135C + 0.0003
Correlation Coefficient		0.9983
Enhancement Factor		71.2
Linear Range	µg/L	0.2–2.5
Linear Range without enrichment	µg/L	15–125
LOD	3s (µg/L)	0.054
LOQ	10s (µg/L)	0.18
Precision	RSD (%) [0.5 µg/L] n = 7	2.30

**Table 4 molecules-29-04189-t004:** Determination of lead in certified material (n = 4).

Sample	Certified (μg/L)	Found (μg/L)	Recovery %
NRC-AQUA-1 Drinking Water	1.37 ± 0.09	1.39 ± 0.12	101.1

**Table 5 molecules-29-04189-t005:** Determination of Pb(II) in real water samples (n = 4).

Tap Water (Beytepe—Ankara)			Seawater (Marmara Sea—Istanbul)			Wastewater (Big Scale WWTP—Ankara)		
Added (μg/L)	Found (μg/L)	Efficiency (%)	Added (μg/L)	Found (μg/L)	Efficiency (%)	Added (μg/L)	Found (μg/L)	Efficiency (%)
	* BDL	-		* BDL	-		* BDL	-
1.00	1.01	101.0	1.00	0.98	98.0	1.00	1.03	103.0
2.00	2.03	101.5	2.00	1.99	99.5	2.00	2.05	102.5

* BDL: below detection limit.

**Table 6 molecules-29-04189-t006:** Furnace heating program for lead analysis.

Stage	Temperature (°C)	Ramp Time (s)	Hold Time (s)	Gas Flow (mL/min)
Drying 1	110	1	20	250
Drying 2	130	15	20	250
Pyrolysis	1000	10	10	250
Atomization ^a^	1500	0	3	0
Clean-Out	1800/2450 ^b^	1	3	250

^a^ Reading step. ^b^ Set for reference and real samples.

**Table 7 molecules-29-04189-t007:** Comparison of proposed IL-SFODME with the literature for the determination of lead.

Preconcentration Method	Instrument	Enhancement Factor	Limit of Detection (µg/L)	Reference
SPE	FAAS	20	3.7	[29]
SPE	FAAS	200	0.4	[30]
DLLME	FAAS	70	2.6	[31]
UAµE-DES	FAAS	71.6	0.66	[32]
UA-SS-LPME	FAAS	101.6	0.63	[33]
CPE	ETAAS	50	0.08	[34]
DLLME	ETAAS	78	0.039	[35]
LPME	ETAAS	52	0.2	[36]
DLLME	ETAAS	-	0.08	[37]
LLE	ICP-MS	23.3	0.011	[38]
IL-SFODME	ETAAS	69.2	0.075	This work

SPE: Solid phase extraction, DLLME: dispersive liquid–liquid microextraction, UAµE-DES: ultrasonic assisted microextraction method based on deep eutectic solvent, UA-SS-LPME: Ultrasonic assisted switchable solvent based on liquid phase microextraction, CPE: Cloud point extraction, LPME: liquid phase microextraction, LLE: liquid–liquid extraction, IL-SFODME: ionic-liquid solidified floating organic drop microextraction.

## Data Availability

The data supporting the findings of this study are not publicly available due to privacy concerns. Requests for access to the data should be directed to the corresponding author, subject to reasonable requests and in compliance with the applicable regulations.

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
