# Peer review of "Ionic Liquid—Solidified Floating Organic Drop Microextraction for the Preconcentration of Lead in Environmental Water Samples Prior to Its Determination with Electrothermal Atomic Absorption Spectrometry"

_molecules, 2024, doi:10.3390/molecules29174189_

Round 1

Reviewer 1 Report

Comments and Suggestions for Authors

In the paper Ionic Liquid - Solidified Floating Organic Drop Microextraction for the Preconcentration of Lead in Environmental Water Samples Prior to Its Determination with Electrothermal Atomic Absorption Spectrometry, authors report an interesting study about the determination of Lead in water by ion liquid extraction method. 

Paper is interestingly, however some parts must be emprived befor publication. 

Introduction

I suggest tonintroduce that metals can be originated by natural or anthropogenic processes. In this context add references such as (Environmental Science: Processes & Impacts 18 (3), 323-329, 2016 / Microchemical Journal 130, 229-235). 

Please, consider that every metal not are biodegradable, in this context i suggest you not use the word non-biodegradable

2.3. Procedure

In the procedure authors write that samplee were diluted to 175 µL. 

I remmeber that describe autodamples use a containers of 2.5 mL and 175 uL is out of sampling of autosampler. 

Moreover, considering that water and ethanol have different density, some errors can be introduced. 

Table 5 

Change Recovery with Accuracy. 

Comments on the Quality of English Language

English Is fine

Author Response

For research article: Ionic Liquid - Solidified Floating Organic Drop Microextraction for the Preconcentration of Lead in Environmental Water Samples Prior to Its Determination with Electrothermal Atomic Absorption Spectrometry

Response to Reviewer 1 Comments

Thank you very much for taking the time to review our manuscript. Please find the detailed responses below and the corresponding revisions/corrections highlighted in the re-submitted files.

Point-by-point response to Comments and Suggestions for Authors

Comments 1: Introduction

I suggest to introduce that metals can be originated by natural or anthropogenic processes. In this context add references such as (Environmental Science: Processes & Impacts 18 (3), 323-329, 2016 / Microchemical Journal 130, 229-235).

Response 1: Thank you for pointing this out. We agree with this comment. Therefore, we have added the proposed references to “Introduction” section and also “references” section.

Comments 2: Please, consider that every metal not are biodegradable, in this context i suggest you not use the word non-biodegradable

Response 2: Agree. We have removed the term “non-biodegradable” in the manuscript. We have, accordingly, changed the related sentences as “Heavy metals, as inorganic pollutants, are particularly concerning because they cannot biodegrade and tend to accumulate in living organisms over time [7].” in section “Introduction”.

Comments 3:  

2.3. Procedure

In the procedure authors write that samplee were diluted to 175 µL.

I remmeber that describe autodamples use a containers of 2.5 mL and 175 uL is out of sampling of autosampler.

Response 3: Reviewer is right.  175 uL is so small for the 2.5 ml containers. Anyway, the experiments were carried out with 1.2 ml cups of containers that gradually narrow downwards. The images of used cups are shown below

Comments 4: Table 5

Change Recovery with Accuracy.

Response 4: In Table 5, the results are expressing the level of accuracy of proposed method. However, since the certified value is certain and accepted value result was expressed as recovery percent and this is the general accepted term for these kind of certified material analysis. Therefore, the recovery term was remained in table.

Reviewer 2 Report

Comments and Suggestions for Authors

The work titled as “Ionic Liquid - Solidified Floating Organic Drop Microextraction for the Preconcentration of Lead in Environmental Water Samples Prior to Its Determination with Electrothermal Atomic Absorption Spectrometry” presents innovation regarding the sample preparation method for lead determination, using ionic liquid and organic extracting solvent as extractants, without using complexing ligands. In addition, the manuscript is well-founded through experiments and conducive results. However, there are some points to be considered by the authors.

-              Keywords: IL-SFODME should be included in the keywords.

-              In the introduction: line 59-60: the authors make the following statement:

"................................ETAAS is preferred due to advantages over conventional tech-59 niques in terms of speed, simplicity, .............."

however, ETAAS does not surpass the other aforementioned techniques (FAAS and ICP-OES) in analysis speed or simplicity of operation......REVIEW THIS STATEMENT....

-              Reagents and Materials: line 109-110: mention the concentration of the lead working solution..... ..it is only found in Figure 1.

-              Figure 2 (caption): How was this final volume obtained? Since the sum of the volumes used in the extraction totals 15.61 mL...

-              Figure 5 (caption) – section “Optimization of extraction time” - Why was 2 mL of buffer solution used in this study? Since the optimal volume of buffer solution was 0.3 mL........

-              Section 3.1.6 “Optimization of extraction temperature” : The experiments carried out in heated environments (temperatures above room temperature) were carried out in what way to ensure these temperatures? Sealed flasks? Hot plate? Make this clear in the manuscript.

-              Section 3.1.9 “Optimization of final volume”: review the discussion of the results presented in the manuscript.........across all the parameters studied, the final volume was 200 µL, but in the study presented for this parameter, the optimal final volume was 175 µL?...the authors do not make this clear in the text. As the parameters that influence SFODME were optimized in a univariate manner, it is worth the authors presenting them in the order in which they were executed or making clear the conditions used in each experiment, because although it is presented in the caption of each figure, there are some figures with incorrect conditions, such as the final volume.......

-              Section 3.3 “Analytical performance of proposed method”: What is the linear range for the curve without enrichment step? present in the manuscript..........

-              Table 7: add a caption to the acronyms of the preconcentration methods (column 1 of Table 7).

Comments on the Quality of English Language

A brief review of English should be done.

Author Response

For research article: Ionic Liquid - Solidified Floating Organic Drop Microextraction for the Preconcentration of Lead in Environmental Water Samples Prior to Its Determination with Electrothermal Atomic Absorption Spectrometry

Response to Reviewer 2 Comments

Thank you very much for taking the time to review our manuscript. Please find the detailed responses below and the corresponding revisions/corrections highlighted in the re-submitted files. We would like to thank the reviewer once again for his/her meticulous review, valuable criticisms that made the manuscript better, and his/her valuable contributions.

Point-by-point response to Comments and Suggestions for Authors

Comment 1:  Keywords: IL-SFODME should be included in the keywords.

Response 1: Thank you for pointing out this. The keyword “IL-SFODME” was added to manuscript and highlighted.

Comment 2:   In the introduction: line 59-60: the authors make the following statement:

"................................ETAAS is preferred due to advantages over conventional tech-59 niques in terms of speed, simplicity, .............."

however, ETAAS does not surpass the other aforementioned techniques (FAAS and ICP-OES) in analysis speed or simplicity of operation......REVIEW THIS STATEMENT....

Response 2: Thank you for warning about this point. Reviewer is absolutely right. ET-AAS is not speed and simple over the compared ones. Since, the terms “speed and simplicity” were removed from the sentences and rewritten the mentioned sentence.

Comment 3:  Reagents and Materials: line 109-110: mention the concentration of the lead working solution..... ..it is only found in Figure 1.

 Response 3: In Figure 1, the mentioned concentration is the one which is used in optimizing the experimental parameters. In the section “Reagents and Materials: line 109-110” mentioned concentrations are the ones used in calibration process in instrument. There are 6 different calibration solutions prepared in 6 different concentrations in increasing amount in order to draw a good calibration graph. The details of working standard solutions which are used in calibration process and calibration standart solution concentrations are given below. It has not been added to the body of the manuscript in order not to lengthen the manuscript unnecessarily, but if the reviewer considers it useful, appropriate information will be added immediately.

Linear Range without enrichment

µg/L

15-125

Concentrations of standard solutions

Pb concentration (ug/L)

15

20

30

50

100

125

Comment 4:  Figure 2 (caption): How was this final volume obtained? Since the sum of the volumes used in the extraction totals 15.61 mL...

Response 4: The final volume of the working solution is 15 mL. Before preparing the solution, a calculation is made. It is calculated that the amount taken from the stock solution will be satisfy the 0.75 ppb of lead ions in the working solution, which will have a final volume of 15 mL, and that amount of lead stock solution is added to the working solution container. Then, other reagents are added and the final solution is completed by adding deionized water to make the final volume 15 mL. An example calculation is given below.

M1 x V1  =  M2 x V2

1.5 ppb x V1  =  0.75 ppb x 15 mL

V1 = 7.5 mL

According to this calculation 7.5 mL of 1.5 ppb stock solution is taken and transferred to the working beaker, and then the other chemicals are added, finally the total volume is adjusted to 15 mL in a scaled beaker by adding proper amount of deionized water.

Comment 5:   Figure 5 (caption) – section “Optimization of extraction time” - Why was 2 mL of buffer solution used in this study? Since the optimal volume of buffer solution was 0.3 mL........

Response 5: Thank you for pointing this out. It is an It is an unintentional mistake. The volume is corrected as 0.3 ml since the experiments were performed at that value. It was written by mistake. The correct value is written through the warning of reviewer and highlighted in manuscript.

Comment 6:  Section 3.1.6 “Optimization of extraction temperature” : The experiments carried out in heated environments (temperatures above room temperature) were carried out in what way to ensure these temperatures? Sealed flasks? Hot plate? Make this clear in the manuscript.

Response 5: In section “Instrumentation,  in page: 3, in line :132, the statement : “The sample solutions were heated to the necessary temperature and stirred at the proper speed using a Kudos HS15-26P multi-heater magnetic stirrer” However, in line with the reviwer's opinions, the relevant explanation was added to the temperature scan section and colored.

Comment 7:  Section 3.1.9 “Optimization of final volume”: review the discussion of the results presented in the manuscript.........across all the parameters studied, the final volume was 200 µL, but in the study presented for this parameter, the optimal final volume was 175 µL?...the authors do not make this clear in the text. As the parameters that influence SFODME were optimized in a univariate manner, it is worth the authors presenting them in the order in which they were executed or making clear the conditions used in each experiment, because although it is presented in the caption of each figure, there are some figures with incorrect conditions, such as the final volume.......

Response 8: In experimental studies, optimization studies were always carried out by keeping only one parameter variable and the other parameters constant. The reason why the author is confused here is that the “final volume” scan is the last scanned parameter. In each scan, the optimum value of the scanned parameter was found, then this value was applied in the next experiment and the path was continued. Until the last parameter, that is, the “final volume” scan, 200 uL of final volume was always applied, but when the “final volume” parameter was scanned, it was seen that 175 uL was better. After this stage, analytical performance studies, the effect of interference species, real and certified sample studies were carried out with all parameters at their optimum values.

Comment 8:  Section 3.3 “Analytical performance of proposed method”: What is the linear range for the curve without enrichment step? present in the manuscript..........

Response 8: The  linear range used to create calibration graph (without enrichment step) information was added to Table 4 as a new row.

Comment 9:  Table 7: add a caption to the acronyms of the preconcentration methods (column 1 of Table 7).

Response 9: The acronyms of the methods mentioned in column 1 of Table 7 were added in a caption section. Thank you to reviewer for this contribution.  

Comments on the Quality of English Language: A brief review of English should be done.

 Response: The entire article has been completely revised, grammatical errors have been corrected and the English of the article has been improved. The changes have been highlighted.

Round 2

Reviewer 1 Report

Comments and Suggestions for Authors

All corrections have been made. However, references must be reported as indicate in journal guideline. 

Comments on the Quality of English Language

English Is fine